
# Acoustic–gravity waves and their role in ionosphere–lower thermosphere coupling

Gordana Jovanovic

University of Montenegro, Faculty of Science and Mathematics

Dzordza Vasingtona bb, 81000 Podgorica, Montenegro

**Correspondence:** Gordana Jovanovic (gordanaj@ucg.ac.me)

**Abstract.** The properties of acoustic–gravity waves (AGWs) in the ionospheric D layer and their role in the D layer–lower thermosphere coupling are studied using the dispersion equation and the reflection coefficient. These analytical equations are an elegant tool for evaluating the contribution of upward–propagating acoustic and gravity waves to the dynamics of the lower thermosphere. It was found that infrasound waves with frequencies $\omega > 0.035 s^{-1}$, which propagate almost vertically,

can reach the lower thermosphere. Also, gravity waves with frequencies lower than $\omega < 0.0087 s^{-1}$, with horizontal phase velocities in the range $159 m/s < v_h < 222 m/s$, and horizontal wavelength 115 km$< \lambda_p <$161 km, are important for the lower thermosphere dynamics. These waves can cause temperature rise in the lower thermosphere and have the potential to generate middle–scale traveling ionospheric disturbances (TIDs). The reflection coefficient for AGWs is highly temperature dependent. During maximum solar activity, the temperature of the lower thermosphere can rise several times. This is the situation where

infrasound waves become a prime candidate for the ionospheric D layer–lower thermosphere coupling, since strongly reflected gravity waves remain trapped in the D layer. Knowing the temperatures of the particular atmospheric layers, we can also know the characteristics of AGWs and vice versa.

## 1 Introduction

Acoustic–gravity waves (AGWs) are able to transport energy and momentum between different layers of the atmosphere. Understanding these waves is essential if we want to comprehend the atmosphere as a system where the layers are coupled. The ionosphere is a part of the Earth's atmosphere located between about 60 km and 1000 km where the charged particles significantly influence its physical and chemical properties (Mitra , 1974; Bothmer and Daglis , 2007). Knowledge of typical AGW characteristics in the ionosphere is important for modelling of coupling between the ionized and neutral atmosphere. Iono-

sphere is constantly exposed to various influences from outer space as well as from the terrestrial atmosphere and lithosphere. The non–periodic and sudden events, such as solar flares (Singh et al. , 2014; Nina et al. , 2017; Chum et al. , 2018), coronal mass ejection (Bochev and Dimitrova , 2003; Balan et al. , 2008), solar eclipses (Singh et al. , 2012), supernova explosions





followed by hard X and $\gamma$ radiation (Inan et al. , 2007), lightnings (Voss et al. , 1998), and some processes in the terrestrial lithosphere like volcanic eruptions and earthquakes (Nenovski et al. , 2010; Argunov and Gotovtsev, 2019), induce space and

time varying ionospheric perturbations. These disturbances cause numerous complex physical, chemical and dynamical phenomena in the ionosphere (Rozhnoi , 2012; Hayakawa et al. , 2010) and may directly affect human activities, especially in the telecommunications.

The atmospheric monitoring depends on the altitude of the considered atmospheric layer. The ionospheric D layer at an altitude of about 60 to 90 km, lies below the area being studied by satellite observations and above the region where balloon measure-

ments find their application. Therefore, its monitoring is based on rocket and radar measurements and on the propagation of very low and low–frequency (VLF/LF) radio waves (Nina and Čadež , 2013). In this way, it is possible to observe a large part of the low ionosphere, detect local perturbations, and sudden events.

The ionospheric D layer and lower thermosphere below 140 km, where AGWs with the specific frequencies and wavelengths are detected, are the focus of this article. We considered the conditions for propagation of AGWs in the D ionospheric layer

and their reflection/transmission on the plane boundary between this layer and the lower thermosphere. This is a way to study the coupling between the ionosphere and the lower thermosphere and to analyze the influence of AGWs on thermospheric processes and characteristics.

The article is structured as follows: Section 2 contains the basic theory of AGWs and the derivation of their dispersion equation. Section 3 presents the analytical equation for the AGW reflection coefficient. In Section 4, the propagation of AGWs

through the ionospheric D layer as well as their reflection/transmission properties are analyzed. Discussion and conclusions are displayed in Sections 5 and 6, respectively.

## 2  Basic equations

The D layer is part of the ionosphere, where typical atmosphere models give $n_n \sim 10^{21} m^{-3}$ for the neutral particle density and $n_p \sim 10^8 m^{-3}$ for charged plasma particles, and where electric and magnetic effects play a minor role in the local atmosphere

dynamics. This is why hydrodynamic (HD) equations, rather than magneto–hydrodynamic (MHD) equations, can be used to analyze wave propagation. The standard set of HD equations describes the dynamics of adiabatic processes in a fully ionized hydrogen plasma in the presence of gravity $\boldsymbol{g} = -g\boldsymbol{e_z}$ with constant acceleration $g = 9.81 m/s^2$:

continuity and ideal gas equation

$$\frac{\partial \rho}{\partial t} + \nabla \cdot (\rho \boldsymbol{v}) = 0, \quad p = \rho RT, \tag{1}$$

momentum equation

$$\rho \left( \frac{\partial \boldsymbol{v}}{\partial t} + \boldsymbol{v} \cdot \nabla \boldsymbol{v} \right) = -\nabla p + \rho \boldsymbol{g} \tag{2}$$

and an adiabatic law for a perfect gas

$$\frac{\partial p}{\partial t} + \boldsymbol{v} \cdot \nabla p = \frac{\gamma p}{\rho} \left( \frac{\partial \rho}{\partial t} + \boldsymbol{v} \cdot \nabla \rho \right). \tag{3}$$





Here, $R = R_0/M$ is the individual gas constant for molecules with molar mass M, $R_0 = 8.314 J/mol K$ is the universal gas constant and $\gamma = c_p/c_v = (j+2)/j$ is the ratio of specific heats for gas particle with $j = 5$ degrees of freedom.

## 2.1 Dispersion equation for AGWs

In what follows, we consider waves whose wavelengths are sufficiently small in comparison with the Earth radius $R_E = 6371$
km. Therefore, the plane parallel geometry can be applied in a locally isothermal medium. Under these assumptions, the atmosphere is taken to be vertically stratified, initially in hydrostatic equilibrium, and then perturbed by harmonic waves of small amplitude. This means that Eqs. (1)–(3) can be linearized by taking any physical quantity $\psi(x,y,z,t)$ as a sum of its basic state unperturbed value $\psi_0(z)$ and a small first order perturbation $\delta\psi(x,y,z,t)$, i.e. $\psi(x,y,z,t) = \psi_0(z) + \delta\psi(x,y,z,t)$, where: $\delta\psi(x,y,z,t) = \psi^{'}(z)e^{i(k_x x + k_y y - \omega t)}$, and $|\psi^{'}| \ll |\psi_0|$. Eqs. (1)–(3), linearized with these perturbations, reduce to three equations: one for the unperturbed basic state and two for small perturbations. The unperturbed basic state is descibed by:

$$\frac{d}{dz}\ln\rho_0(z) + \frac{1}{H} = 0, \quad p_0 = \rho_0 R T_0, \quad \text{with} \quad T_0 = const,$$

whose solution is:

$$\rho_0(z) = \rho_0(0)e^{-z/H} \quad \text{or} \quad p_0(z) = p_0(0)e^{-z/H}, \tag{4}$$

where $H = p_0(0)/\rho_0(0) = v_s^2/\gamma g = const$ is the characteristic scale–height of the isothermal atmosphere.

The small perturbations are governed by equations (Jovanovic , 2016):

$$\frac{d\xi_z^{'}}{dz} = C_1\xi_z^{'} - C_2 p^{'}, \quad \frac{dp^{'}}{dz} - g\frac{d\rho_0}{dz}\xi_z^{'} = C_3\xi_z^{'} - C_1 p^{'}, \tag{5}$$

where $\xi_z^{'} = iv_z^{'}/\omega$ is the z–component (i.e. the vertical component) of the fluid displacement, while $p^{'}$ is the pressure perturbation. The coefficients in Eqs. (5) are:

$$C_1 = \frac{g}{v_s^2}, \quad C_2 = \frac{\omega^2 - k_p^2 v_s^2}{\rho_0(z)v_s^2\omega^2}, \quad C_3 = \rho_0(z)\left(\omega^2 + \frac{g^2}{v_s^2}\right). \tag{6}$$

The density distribution $\rho_0(z)$ is given by Eq. (4) and $k_p^2 = k_x^2 + k_y^2$ designates square of the horizontal wavenumber. The Eqs. (5)–(6) allow the following solutions for the vertical displacement $\xi_z^{'}$ and the pressure perturbation $p^{'}$:

$$\xi_z^{'}(z) = \xi_z^{'}(0)e^{\frac{z}{2H}}e^{ik_z z}, \quad p^{'}(z) = p^{'}(0)e^{\frac{-z}{2H}}e^{ik_z z}. \tag{7}$$

Eqs. (5) with solutions Eqs. (7) yield the dispersion equation for AGWs:

$$k_z^2 = \frac{\omega^2(\omega^2 - \omega_{co}^2) - k_p^2 v_s^2(\omega^2 - \omega_{BV}^2)}{\omega^2 v_s^2}. \tag{8}$$

Here, $k_z$ is the vertical wavenumber, $\omega_{co}^2 = \gamma^2 g^2/4v_s^2 = v_s^2/4H^2$ is the square of the acoustic wave cutoff frequency, and $\omega_{BV}^2 = (\gamma - 1)g^2/v_s^2$ is the square of the Brunt–Väisälää frequency. This equation is quadratic in $\omega^2$ which indicates the existence of two wave modes in the considered stratified atmosphere: the acoustic and gravity modes. Stratification in a vertical




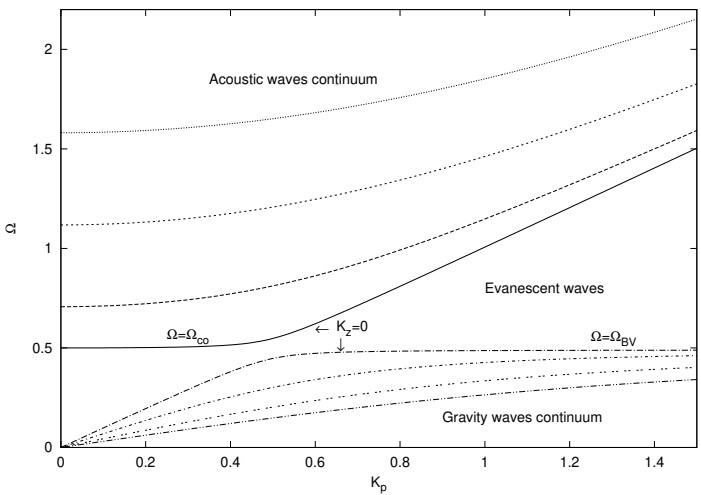

**Figure 1.** Dispersion curves for AGWs. Two sets of curves are related to acoustic and gravity waves, which cannot propagate below the acoustic cutoff frequency $\Omega_{co} = \omega_{co}H/v_s$ and above the Brunt–Väisälää frequency $\Omega_{BV} = \omega_{BV}H/v_s$, respectively.

direction, caused by gravity and given by Eq. (4), introduces cutoff frequencies–acoustic cutoff frequency below which acoustic waves cannot propagate and Brunt–Väisälää frequency above which gravity waves cannot propagate. Therefore, the branches of acoustic and gravity waves are present. Between them are evanescent waves that do not propagate, Fig. 1. The dispersion equation Eq. (8) can be expressed in terms of wavelengths and wave frequencies $\omega$ in the following way:

$$\lambda_z^2(\omega) = \frac{A_2(\omega)\lambda_p^2}{\lambda_p^2 - A_0(\omega)}, \tag{9}$$

where

$$A_0(\omega) = \frac{4\pi^2 v_s^2(\omega^2 - \omega_{BV}^2)}{\omega^2(\omega^2 - \omega_{co}^2)}, \qquad A_2(\omega) = \frac{4\pi^2 v_s^2}{\omega^2 - \omega_{co}^2}.$$

This equation will be useful for further analysis.

The physical quantities in the dispersion equation can be made dimensionless by appropriate scalings: $K_p = k_p H$, $K_z = k_z H$, $\Omega = \omega H/v_s$, $\Omega_{co} = \omega_{co}H/v_s = 0.5$ and $\Omega_{BV} = \omega_{BV}H/v_s = \sqrt{\gamma-1}/\gamma = 0.45$. Now, the dispersion equation Eq. (8) has the dimensionless form:

$$K_z^2 = \Omega^2 - \Omega_{co}^2 - \frac{K_p^2(\Omega^2 - \Omega_{BV}^2)}{\Omega^2}. \tag{10}$$

The AGWs propagate in the vertical direction if $K_z^2 > 0$. This is fulfilled when

$$K_p^2 < \frac{\Omega^2(\Omega^2 - \Omega_{co}^2)}{\Omega^2 - \Omega_{BV}^2}, \tag{11}$$

i.e. when dimensionless horizontal phase velocity is

$$V_h^2 = \frac{\Omega^2}{K_p^2} > \frac{\Omega^2 - \Omega_{BV}^2}{\Omega^2 - \Omega_{co}^2}. \tag{12}$$





The AGWs become evanescent when $K_p^2 > \frac{\Omega^2(\Omega^2 - \Omega_{co}^2)}{\Omega^2 - \Omega_{BV}^2}$ and $V_h^2 < \frac{\Omega^2 - \Omega_{BV}^2}{\Omega^2 - \Omega_{co}^2}$. The boundary between propagating and evanescent regions is given by $K_z = 0$. Acoustic waves with frequencies close to the acoustic cutoff frequency $\Omega \approx \Omega_{co} = 0.5$, are more influenced by gravity than those with high frequencies, when $\Omega \gg \Omega_{co}$. Hence, gravity–modified acoustic waves and pure acoustic waves coexist in the stratified atmosphere (Mihalas , 1984). The Eq. (10) shows that the vertical wavenumber $K_z$ has a maximum value for $K_p = 0$, i.e.

$$K_{zmax} = K = \sqrt{\Omega^2 - \Omega_{co}^2}. \tag{13}$$

This equation describes acoustic waves that propagate only in the vertical direction.

Gravity waves, in contrast to acoustic waves, are not able to travel vertically with $K_p = 0$, which means there are no pure vertically propagating gravity waves (Mihalas , 1984). Therefore, they propagate obliquely through the stratified atmosphere in accordance with Eq. (10). For the very low frequencies, when $\Omega \ll \Omega_{BV} = 0.45$, gravity waves propagate with:

$$K_z \approx \frac{K_p \Omega_{BV}}{\Omega}, \quad i.e., \quad \frac{\lambda_p}{\lambda_z} \approx \frac{\omega_{BV}}{\omega}. \tag{14}$$

Dimensionless equations are used because they are valid in each stratified medium, like Earth's, planets or the solar atmosphere. When we rewrite them using characteristic frequencies and temperatures, we obtain the equations for particular atmospheric layers as will be done in Section 4 for the ionospheric D layer.

## 3   Reflection coefficient of AGWs

The considered basic state in the stratified atmosphere is composed of two half–spaces with constant sound speeds, separated by a horizontal plane boundary $z = 0$. The two regions are characterized by the corresponding plasma densities $\rho_{01}$ and $\rho_{02}$ adjacent to the lower and upper side of the boundary $z = 0$. The unperturbed density profile can be expressed as follows:

$$\rho_0(z) = \rho_{01} e^{-z/H_1}, \; z < 0, \text{ region (1)}, \quad \rho_0(z) = \rho_{02} e^{-z/H_2}, \; z > 0, \text{ region (2)}, \tag{15}$$

where $H(n) = v_{sn}^2/\gamma g$, $n = 1, 2$. There is a density, pressure, and temperature jump across $z = 0$. The boundary condition that has to be applied at $z = 0$ in the basic state is the continuity of the unperturbed pressure $p_0$ at $z = 0$, (Jovanovic , 2016), which yield:

$$\frac{\rho_{02}}{\rho_{01}} = \frac{v_{s1}^2}{v_{s2}^2} = \frac{T_1}{T_2} = s = const. \tag{16}$$

The boundary conditions for perturbations are continuity of both the vertical fluid displacement $\xi_z^{'}$ and the pressure perturbation $p^{'} - g\rho_0(z)\xi_z^{'}$ at the boundary $z = 0$. Also, the energy density of the perturbations has to diminish to zero as $|z|$ tends to infinity. The harmonic wave, which propagates through regions (1) and (2), does not change its frequency, and the horizontal wavevector component $K_p$, parallel to the boundary $z = 0$. However, the vertical wavevector component $K_z$ has a discontinuity at the boundary $z = 0$, where it changes from $K_{z1}$ to $K_{z2}$ according to the dispersion equation Eq. (10). We assume that a wave propagates from the lower region (1) upward towards the boundary $z = 0$, and that the waves continuing past it are absorbed



with no reflection in the upper region (2). In this case, in the lower region, the perturbations are the superposition of the incident and reflected waves, while in the upper region, there is only the transmitted wave. The reflection coefficient of AGWs is defined as the square of the absolute value of the reflection amplitude. Using dimensionless physical values for brevity, the reflection coefficient can be written as (see details in Jovanović , 2014):

$$R = \left[ \frac{\left[ \left(1-\frac{\gamma}{2}\right)\left(\frac{1}{V_h^2-1} - \frac{s^2}{sV_h^2-1}\right) + \frac{(s-1)}{V_h^2} \right]^2 + \frac{\gamma^2\Omega^2}{V_{v1}^2}\left(\frac{V_{v1}^2}{V_{v2}^2} \cdot \frac{s^2}{(sV_h^2-1)^2} - \frac{1}{(V_h^2-1)^2}\right)}{\left[ \left(1-\frac{\gamma}{2}\right)\left(\frac{1}{V_h^2-1} - \frac{s^2}{sV_h^2-1}\right) + \frac{(s-1)}{V_h^2} \right]^2 + \frac{\gamma^2\Omega^2}{V_{v1}^2}\left[ \frac{V_{v1}}{V_{v2}} \cdot \frac{s}{sV_h^2-1} + \frac{1}{V_h^2-1} \right]^2} \right]^2 +$$

$$\left[ \frac{\frac{2\gamma\Omega}{V_{v1}(V_h^2-1)}\left[ \left(1-\frac{\gamma}{2}\right)\left(\frac{1}{V_h^2-1} - \frac{s^2}{sV_h^2-1}\right) + \frac{(s-1)}{V_h^2} \right]}{\left[ \left(1-\frac{\gamma}{2}\right)\left(\frac{1}{V_h^2-1} - \frac{s^2}{sV_h^2-1}\right) + \frac{(s-1)}{V_h^2} \right]^2 + \frac{\gamma^2\Omega^2}{V_{v1}^2}\left[ \frac{V_{v1}}{V_{v2}} \cdot \frac{s}{sV_h^2-1} + \frac{1}{V_h^2-1} \right]^2} \right]^2 . \tag{17}$$

Here, $V_{v1}$ and $V_{v2}$ are the vertical phase velocities of AGWs in regions (1) and (2) respectively, given by the equations:

$$V_{v1} = \frac{\Omega}{K_{z1}} = \frac{V_h\Omega}{\sqrt{V_h^2(\Omega^2 - \Omega_{co}^2) - (\Omega^2 - \Omega_{BV}^2)}}, \tag{18}$$

and

$$V_{v2} = \frac{\Omega}{K_{z2}} = \frac{V_h\Omega}{\sqrt{sV_h^2(\Omega^2 - s\Omega_{co}^2) - (\Omega^2 - s\Omega_{BV}^2)}}, \tag{19}$$

while $V_h$ is horizontal phase velocity given by Eq. (12). If $V_{v1}^2$ and $V_{v2}^2$ are positive, AGWs propagate through regions (1) and (2), respectively. If $V_{v1}^2, V_{v2}^2 < 0$, these waves are evanescent and not of interest to this study.

## 4 Results

In this Section, the analytical equations derived in Sections 2 and 3 are used to analyze the propagation of AGWs and their reflection/transmission properties in the ionospheric D layer.

### 4.1 AGWs in the ionospheric D layer

Acoustic–gravity waves which propagate in the lower ionosphere below 90 km can be generated from below, where hydrodynamic motions can be induced by atmospheric convective motions (Sindelarova, Buresova, and Chum , 2009), in the lithosphere (Nina et al. 2021; Boudjada et al. , 2024), and from above, due to sunrise and sunset effects (Afraimovich et al. , 2009; Nina and Čadež , 2013; Nina et al. , 2017). These perturbations may result into various patterns of either eigenmodes or driven linear waves in the atmosphere. The focus of this research is on the driven AGWs and their role in the ionosphere and the lower thermosphere coupling. Therefore, propagation of AGWs in the vertical direction is particularly important. For the considered isothermal ionospheric D layer with a temperature $T = 250$ K and $\gamma = 1.4$, sound velocity is $v_s = \sqrt{\gamma R T} = 317 m/s$, and $H = 7317$ m. This is in accordance with Lizunov and Hayakawa (2004). For a gravity modified acoustic wave with frequency

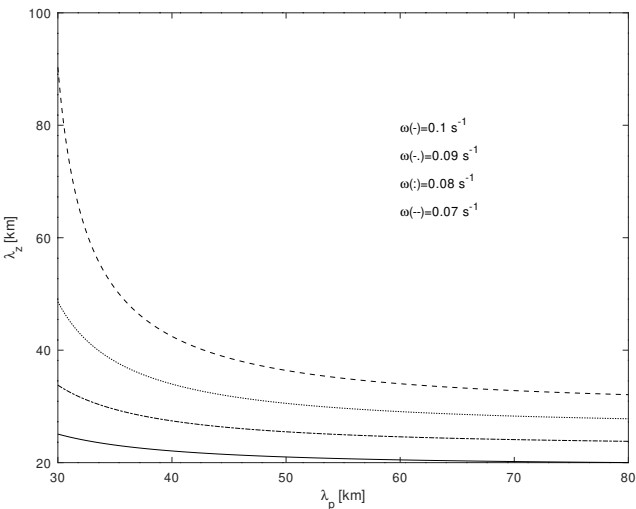

**Figure 2.** Vertical wavelength $\lambda_z$ of acoustic waves from dispersion equation (9) as a function of horizontal wavelength $\lambda_p$, for given frequencies $\omega > \omega_{co} = 0.021 s^{-1}$.

near acoustic cutoff frequency $\omega = 0.022 s^{-1} \geq \omega_{co} = 0.021 s^{-1}$, Eq. (13) enables the calculation of $\lambda_z \geq \lambda_{zmin} \approx 460$ km. For the pure acoustic wave, with frequency much greater than acoustic cutoff frequency, $\omega \gg \omega_{co}$, i.e. $\omega = 10 \cdot \omega_{co} = 0.21 s^{-1}$, this value is $\lambda_z \geq \lambda_{zmin} \approx 9.2$ km. It can be noticed that gravity–modified acoustic waves have much longer vertical wave-

lengths than pure acoustic waves. Therefore, acoustic waves with frequencies near the acoustic cutoff frequency $\omega_{co}$ have the best chance for vertical propagation through the ionospheric D layer towards the lower thermosphere. Acoustic waves in Fig. 2 are detected in the ionosperic D layer using VLF waves (Nina and Čadež , 2013).

Gravity waves with high frequencies, $\omega \approx \omega_{BV}$, and with low frequencies, $\omega \ll \omega_{BV}$, are presented in Figs. 3 and 4, respectively. The Eq. (14) shows that low–frequency gravity waves have much longer horizontal than vertical wavelengths, i.e. they

propagate more horizontally than vertically, Fig. 4. In addition, for a given $\lambda_p$, the vertical wavelengths of low–frequency gravity waves are shorter than those of gravity waves with a frequency that is close to the Brunt–Väisälää frequency, Figs. 3 and 4. The vertical phase velocities of these waves are smaller than those of high–frequency gravity waves. Therefore, high–frequency gravity waves propagate faster upward through the ionospheric D layer towards the lower thermosphere. Figures 3 and 4 show gravity waves that were found in the ionospheric D layer (Nina and Čadež , 2013). They can be induced in situ at sunrise and

sunset due to motions of the solar terminator. Low–frequency gravity waves are observed near OH layer at an altitude of about 87 km and near $O_2$ layer at an altitude of about 94 km by the mesospheric temperature mapper (Yuan et al. , 2016). Their frequencies are $\omega = 0.0011 s^{-1}$ and $\omega = 0.0014 s^{-1}$, respectively, and are even lower than the gravity wave frequencies in Fig. 4.

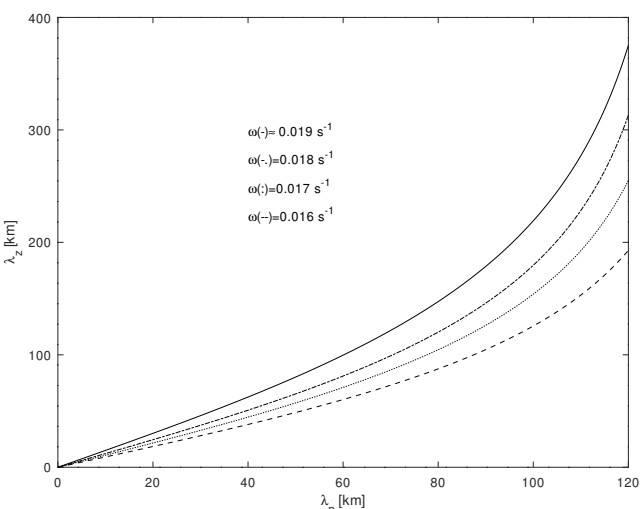

**Figure 3.** Vertical wavelength $\lambda_z$ of gravity waves from dispersion Eq. (9) as a function of horizontal wavelength $\lambda_p$, for given frequencies $\omega \approx \omega_{BV} = 0.0195 s^{-1}$.

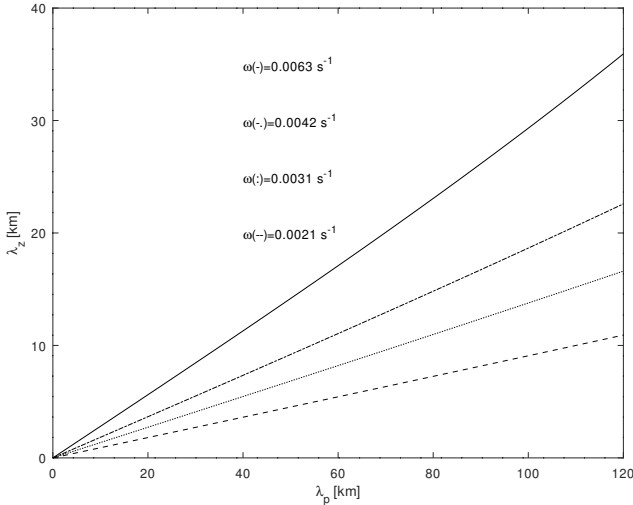

**Figure 4.** Vertical wavelength $\lambda_z$ of gravity waves from dispersion Eq. (9) as a function of horizontal wavelength $\lambda_p$, for given frequencies $\omega \ll \omega_{BV} = 0.0195 s^{-1}$.





## 4.2 Reflection coefficient of AGWs at the D layer–lower thermosphere boundary

We assume that $z = 0$, Eq. (15), is the plane boundary between the ionospheric D layer at an altitude of $60 - 90$ km, region (1), and lower thermosphere at an altitude of $90$ to about $140$ km, region (2). At this boundary, AGWs coming from below can be reflected in the D layer or transmitted into the lower thermosphere. The temperature of the D layer is $T_1 = 250$ K, while the temperature of the lower thermosphere is $T_2 = 500$ K. Therefore, Eq. (16) gives $s = 0.5$. The reflection coefficients of acoustic and gravity waves will be analyzed separately.

### 4.2.1 Reflection coefficient of the acoustic waves

Figure 5 shows the reflection coefficient as a function of frequency $\Omega$, for acoustic waves at the D layer–lower thermosphere plane boundary $z = 0$, when $s = 0.5$. Acoustic waves in the frequency range $\Omega_{co} < \Omega < 0.8$ are reflected at this boundary to a somewhat greater extent. The reflection coefficient strongly decreases with increasing frequency, and acoustic waves with frequencies $\Omega > 0.8$, i.e. $\omega = 0.035 s^{-1}$, can easily propagate through the D layer–lower thermosphere boundary. These waves

could affect the thermospheric temperature and dynamics by depositing their momentum and energy in the lower thermosphere. The value of the horizontal phase velocity $V_h$ does not significantly affect the reflection coefficient, except in the case when $V_h = 1/\sqrt{s} = 1.41$, i.e. for the horizontal phase velocity of the acoustic waves $v_{ph} = 1.41 v_s = 447 m/s$, when total internal reflection occurs. Waves with this horizontal velocity cannot penetrate the thermosphere. Acoustic waves with horizontal phase velocities $V_h > 1.41$ can propagate through the D layer–lower thermosphere boundary and extend further into the ther-

mosphere, especially if $\Omega > 1$. Their reflection coefficient slowly decreases with the increase of $V_h$ for a given frequency $\Omega$.

### 4.2.2 Reflection coefficient of the gravity waves

The reflection coefficient for gravity waves increases when the frequency $\Omega$ increases and decreases with increasing horizontal phase velocity $V_h$ for a given frequency $\Omega$, Fig. 6. These waves can propagate in both regions, in the ionospheric D layer and

in the lower thermosphere, if their frequencies are lower than the cutoff frequency $\Omega = \sqrt{s}\Omega_{BV} = 0.32$, or $\omega = 0.014 s^{-1}$, and their horizontal phase velocities are lower than $V_h = \Omega_{BV}/\Omega_{co} = 0.9$, i.e. $v_h = 0.9 v_s = 285 \, m/s$. Gravity waves with frequencies much lower than the Brunt–Väisälää frequency and with high horizontal phase velocities are candidates for crossing the D layer–lower thermosphere boundary. Contrary to this, gravity waves with frequencies near the cutoff frequency $\Omega = \sqrt{s}\Omega_{BV}$ are strongly reflected at the D layer–lower thermosphere boundary. For the horizontal phase velocity $V_h = 0.9$, total internal

reflection occurs, and the reflection coefficient is equal to unity.

## 5 Discussion

It is known that high–frequency acoustic waves are strongly absorbed by the atmosphere (Sindelarova, Buresova, and Chum , 2009). The rate of absorption is proportional to the wave frequency squared. Therefore, only acoustic waves with the low

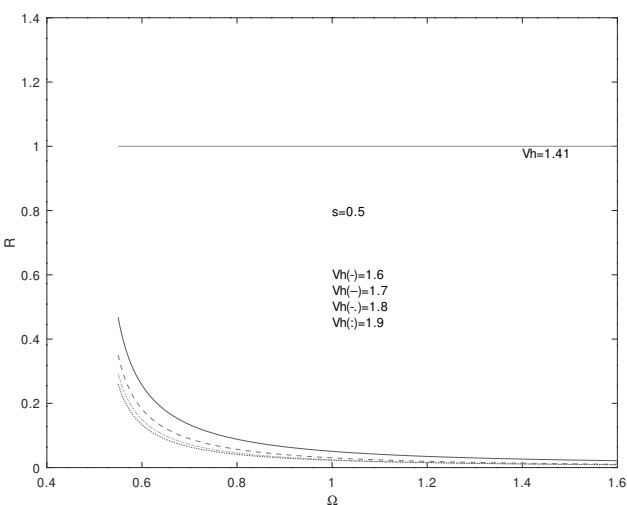

**Figure 5.** Reflection coefficient for acoustic waves at the D layer–lower thermosphere plane boundary $z = 0$, as a function of frequency $\Omega$ and parameter $s = 0.5$. If $V_h = 1/\sqrt{s} = 1.41$, the reflection coefficient is $R = 1$ and total internal reflection occurs.

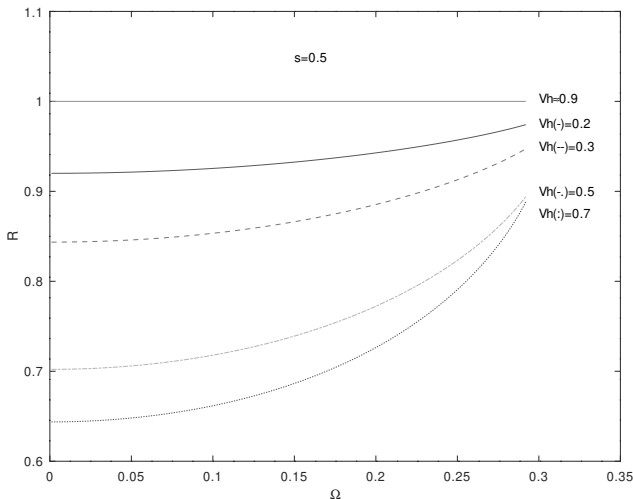

**Figure 6.** Reflection coefficient for gravity waves at the D layer–lower thermosphere plane boundary $z = 0$, as a function of frequency $\Omega$ and parameter $s = 0.5$. If $V_h = \Omega_{BV}/\Omega_{co} = 0.9$, the reflection coefficient is $R = 1$ and total internal reflection occurs.





frequencies (infrasound) may propagate through ionospheric D layer and eventually through the lower thermosphere. Indeed,
it was found that only acoustic waves with periods less than 4 minutes, i.e. $\Omega > 0.6$, or $\omega > 0.026 s^{-1}$ propagating almost
vertically are able to reach the lower thermosphere (Blanc , 1985; Schulthess , 2022). In Fig. 5, the reflection coefficient
for infrasound waves is presented since the dimensionless frequency $\Omega = 1.6$ corresponds to frequency $\omega = 0.069 s^{-1}$, i.e.
$\nu = \omega/2\pi = 0.01$ Hz. These waves, with horizontal phase velocities $v_h > 447$ $m/s$ and with minimum vertical phase veloc-
ity $v_{vmin} > 317$ $m/s$, have the best chance of reaching the thermosphere if they propagate almost vertically with infrasound
frequencies $\omega > 0.035 s^{-1}$. Although infrasound waves dissipate their energy in the lower thermosphere, they are not the first
option for raising its temperature. Namely, the influence of acoustic wave energetics into the ionosphere/lower thermosphere
is weak (Lizunov and Hayakawa , 2004). It appears that the temperature in the thermosphere is increased by low–frequency
gravity waves coming from below (Sindelarova, Buresova, and Chum , 2009). Their reflection coefficient for the ionospheric
D layer–lower thermosphere boundary is shown in Fig. 6. Gravity waves with horizontal phase velocities $V_h < 0.5$ are easily
reflected from the boundary between the D layer and lower thermosphere and will likely remain trapped at lower altitudes.
Only waves with horizontal phase velocities $0.5 < V_h < 0.7$, i.e. $159$ $m/s < v_h < 222$ $m/s$, and with low frequencies $\Omega < 0.2$,
or $\omega < 0.0087 s^{-1}$, are important for the dynamics of the middle atmosphere. Horizontal wavelengths for these waves are
in the range of $115$ km$< \lambda_p < 161$ km. This is consistent with the results known from the scientific literature (Fritts et al.
, 2014; Bakhmetieva et al. , 2019), which emphasize that gravity waves with periods as short as 10 minutes (i.e. $\Omega < 0.24$,
or $\omega < 0.01 s^{-1}$ ) can carry significant momentum flux vertically. These waves with wavelengths $\lambda_p \approx 100 - 200$ km, play
an important role in the coupling between the ionospheric D layer and the lower thermosphere. They are responsible for the
generation of middle–scale traveling ionospheric disturbances (TIDs) with periods from 15 minutes to 3 hours, velocities from
100 to 250 $m/s$ and horizontal wavelength of approximately a few hundred kilometers (Lizunov and Hayakawa , 2004). It
seems that they are causing a rise in temperature in the lower thermosphere through the process of gravity wave breaking and
dissipation due to kinematic viscosity and thermal diffusivity (Vadas , 2007; Sindelarova, Buresova, and Chum , 2009; Yuan et
al. , 2016). A similar situation can be found in the solar atmosphere (Fleck et al. , 2021), and at the photosphere–chromosphere
boundary (Marmolino et al. , 1993; Jovanović , 2014), with the parameter $s = 0.6$.

Gravity waves dissipate their energy contributing to local heating of the thermosphere at higher altitudes during extreme solar
minimum, since the kinematic viscosity is much smaller in warmer than in colder thermosphere at the same altitude (Sinde-
larova, Buresova, and Chum , 2009). During extreme solar minimum the lower thermosphere is relatively cold, $T \approx 500$ K,
while during active solar conditions the temperature in thermosphere can be $T \approx 2000$ K (Vadas , 2007). The current $25^{th}$
solar cycle, which began in Decemeber 2019, is expected to have maximum activity in July 2025. This solar activity could
increase the temeperature in the lower thermosphere several times. The reflection coefficient for acoustic waves in active solar
conditions varies with the frequency $\Omega$ and the parameter $s = 250K/2000K = 0.125$, as depicted in Fig. 7. The reflection coef-
ficient decreases in the frequency range $\Omega_{co} < \Omega < 1.5$. Acoustic waves with $V_h \geq 1/\sqrt{s} \approx 2.83$ are the best candidates to pass
through D layer–lower thermosphere boundary and propagate further in the thermosphere. Acoustic waves with $V_h \gg 1/\sqrt{s}$
are the most susceptible to reflection. This is opposite situation compared to the reflection coefficient for acoustic waves with
$s = 0.5$, Fig. 5, where the waves with $V_h \approx 1/\sqrt{s}$ are the ones that are most prone to reflection. For frequencies $\Omega > 1.5$, the

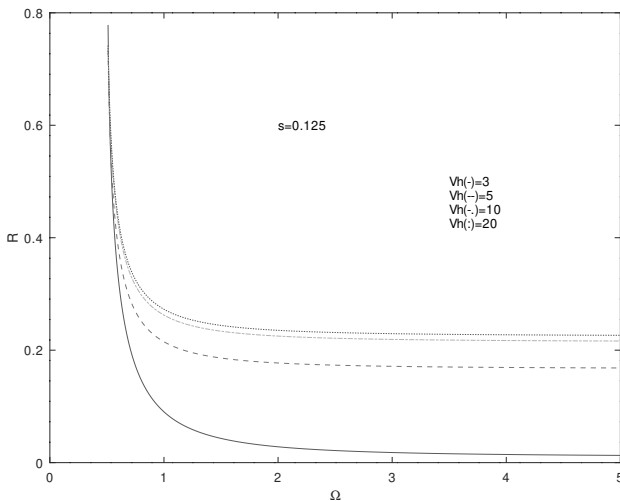

**Figure 7.** Reflection coefficient for acoustic waves at the D layer–lower thermosphere plane boundary $z = 0$, as a function of frequency $\Omega$ and parameter $s = 0.125$.

reflection coefficient decreases very slowly and remains almost constant.

The reflection coefficient for gravity waves in active solar conditions as a function of frequency $\Omega$ and the parameter $s = 0.125$, is shown in Fig. 8. It has very high values for all gravity waves propagating with allowed $V_h < 0.9$. These waves can hardly pass the bounadry between the ionospheric D layer and the lower thermosphere. It seems that they are trapped in the ionospheric D layer and cannot propagate through the thermosphere. Therefore, infrasound could be instrument in coupling between ionospheric D layer and lower thermosphere during solar maximum activity.

The conditions for AGW propagation, as well as their reflection coefficient, strongly depend on the temperature through $v_s$ and parameter $s$, Eqs. (8) and (17). Therefore, any change in temperature can affect the propagation of AGWs and their reflection and transmission features. This means that the detection of these waves depends on the current temperature in the region being observed. A similar situation exists with the detection of AGWs by lidar or any other instrument because their positions relative to the wave source region will determine which AGW characteristic can be observed (Yuan et al. , 2016).

One of the important effects of AGWs and especially gravity waves is their influence on the concentration of charged particles in the ionospheric E layer embedded in the lower thermosphere at an altitude of $90 - 140$ km. Namely, the concentration of charged particles becomes time–dependent in the presence of waves. The changed characteristics of this layer affect the reflection of radio waves and telecommunication connections (Zawdie et al. , 2022). A similar situation is seen in strong natural hazards when earthquakes of magnitude Mv5.5+ are studied by VLF/LF radio waves. A physical interpretation is based on

atmospheric gravity waves which could alter the ionospheric E layer and modulate the height of the VLF/LF waves reflection (Eichelberger et al. , 2024).

An interesting approach to the study of linear AGWs has been made by Cheremnykh (2020). He suggests that AGWs in an





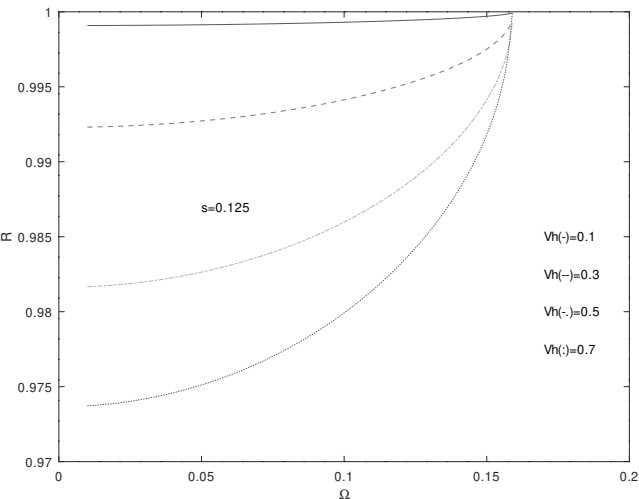

**Figure 8.** Reflection coefficient for gravity waves at the D layer–lower thermosphere plane boundary $z = 0$, as a function of frequency $\Omega$ and parameter $s = 0.125$.

isothermal atmosphere can be considered as a superposition of oscillations that occur simultaneously at two natural frequencies–acoustic and gravitational for a fixed wavelength.

## 6  Conclusions

In this article, analytical equations are used to study AGWs propagation through the ionospheric D layer and the D layer–lower thermosphere coupling. The dispersion equation and the reflection coefficient show that infrasound waves with frequencies $\omega > 0.035 s^{-1}$ that propagate almost vertically can reach the lower thermosphere. Gravity waves propagate in both regions–the ionospheric D layer and the lower thermosphere if their frequencies are $\omega < 0.014 s^{-1}$ and their horizontal phase velocities are $v_h < 285 m/s$. Gravity waves with frequencies much lower than the Brunt–Väisälää frequency $\omega_{BV} = 0.0195 s^{-1}$ propagate more horizontally than vertically because $\lambda_p \ll \lambda_z$. These waves have lower vertical phase velocities than high–frequency gravity waves with $\omega \approx \omega_{BV}$ which travel faster through the ionospheric D layer towards the lower thermosphere. The reflection coefficient is the smallest for the gravity waves with the frequencies $\omega < 0.0087 s^{-1}$, horizontal phase velocities $159 m/s < v_h < 222 m/s$, and horizontal wavelengths 115 km$< \lambda_p <$161 km, which is in accordance with the results known in the scientific literature (Lizunov and Hayakawa , 2004; Fritts et al. , 2014; Bakhmetieva et al. , 2019). These waves can generate the middle–scale TIDs and cause temperature rise in the lower ionosphere.

The reflection coefficient is highly temperature dependent. It changes significantly during the pronounced solar maximum when the temperature in the lower thermosphere can rise several times. A strong increase in the reflection coefficient for gravity waves indicates that they cannot pass D layer–lower thermosphere boundary. Therefore, infrasound waves are better



coupling instruments.

There is broad scientific interest in the future study of AGWs. This is particularly attributed to the study of natural hazards, telecommunications and navigation, and space weather. Due to the complex nature of this process, differences between model results and observations are expected (Klymenko et al. , 2021).


*Author contributions.* There is only one author and the article is the result of the work of one author.

*Competing interests.* The author declares that they have no conflict of interest.

*Acknowledgements.* The research and writing of this work was supported by the Montenegrin National Project "Physics of Ionized Gases and Ionized Radiation".



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
