# Peer review of "Acoustic-gravity waves and their role in the ionospheric D region-lower thermosphere interaction"

_Annales Geophysicae, 2024_

## Referee Comment (RC2)

Annales Geophysicae

**Acoustic-gravity waves and their role in ionosphere-lower thermosphere coupling**

*Gordana Jovanovich*

**Recommendations regarding this manuscript:**

The article needs major revision

**Abstract** reflects the content of the article.

The author analyzed the properties of acoustic–gravity waves (AGWs) in the ionospheric D layer and their role in the D layer–lower thermosphere coupling using the dispersion equation and the reflection coefficient. Such studies are important for understanding the energy exchange between different altitude levels in the atmosphere using AGW.

There are a number of significant comments to the manuscript.

**Comments:**

In my opinion, the title of the work does not fully correspond to its content. The term "ionosphere-lower thermosphere coupling" is usually used in the sense of the interaction of the neutral component of the atmosphere with the ionospheric plasma, which is realized through the collision of particles. In this sense, the analysis of the dispersion equation of AGW and reflection coefficients is not enough for the study of "ionosphere-lower thermosphere coupling". In this work, the D layer is considered formally, simply as a certain altitude level of the atmosphere (60-90 km), and the ionospheric plasma is not involved in this consideration in any way. In fact, only the neutral atmospheric environment is analyzed, which is described by the HD system of equations (1)-(3).

The considered model is questionable. In fact, the work examines two isothermal altitudinal layers with different temperatures, separated by a conventional boundary. The lower layer denotes the D layer, and the upper one denotes lower thermosphere. In the lower considered altitude interval of 60-90 km, the atmosphere is significantly non-isothermal. In particular, these heights include the temperature minimum in the mesopause. In the lower thermosphere (90-140 km) the largest altitudinal temperature gradient in the atmosphere is observed. That is, both the upper and lower layers considered in the work can hardly be considered isothermal. At the same time, the author uses the theory for freely propagating AGWs, developed for an isothermal atmosphere. Based on this theory, the dispersion equation and reflection coefficients were obtained. It is advisable to apply the isothermal theory of AGW in the thermosphere above 200 km, where the temperature almost does not change with height.

P. 4. Lines 46-47. "The standard set of HD equations describes the dynamics of adiabatic processes in a fully ionized hydrogen plasma (?) in the presence of gravity…". The statement is incorrect. In this work, the analysis was carried out on the basis of the linear theory of AGW for a neutral isothermal atmosphere. In fact, the system of equations (1)-(3) for a neutral atmosphere is given below.

P. 4. The analysis of the dispersion equation of AGW (10) given in lines 90-94 is incomplete. In the sense that inequalities (11) and (12) are valid only for the acoustic branch when $\Omega^2 - \Omega_{co}^2 > 0$ and $\Omega^2 - \Omega_{BV}^2 > 0$. However, for the gravitational branch, $\Omega^2 - \Omega_{co}^2 < 0$ and $\Omega^2 - \Omega_{BV}^2 < 0$ (See Fig.1) are always performed. Then, for the condition of free propagation of gravitational waves ($K_z^2 > 0$), the

dispersion equation (10) follows $K_p^2 > \dfrac{\Omega^2\left(\Omega^2 - \Omega_{co}^2\right)}{\Omega^2 - \Omega_{BV}^2}$ and $V_h^2 < \dfrac{\Omega^2 - \Omega_{BV}^2}{\Omega^2 - \Omega_{co}^2}$. That is, for the gravity branch of AGW, inequalities (11) and (12) have the opposite sign. Accordingly, the reflection conditions of the AGW should be recorded separately for the acoustic and gravitational branches. It is necessary to analyze this in the text and how it will affect the results.

P. 5. Line 109. The two regions are characterized by the corresponding plasma densities $\rho 01$ and $\rho 02$. These are the densities of the neutral atmosphere, not the plasma.

P.5. Line 115. In the Earth's atmosphere, the background density decreases with height, and the temperature can have both a positive and a negative gradient. How can the boundary condition for the background parameters (16) be fulfilled with a negative temperature gradient?

P.5. Line 125. The work (Jovanovic, 2014) is not available to me. Based on the inequalities (11) and (12) given by the author, I can assume that the reflection coefficient (17) and expressions (18) and (19) correctly take into account only the acoustic branch. In this regard, it is not clear how the reflection coefficients for the gravitational branch presented in Fig. 6 and 8 were calculated.

I think that the dependence of the vertical wavelength on the horizontal wavelength at different frequencies (see Fig. 2, 3 and 4) is not necessarily shown in the work.

The work needs significant revision, first of all, attention should be paid to the specified differences between the acoustic and gravity branches.

---

## Author Response (AR1)

**Questions and answers**

Dear topic editor,

I include the reviewers' recommendations and suggestions in the second version of the article. Please find it attached. Corrections are marked in red.

Regards!